

# GHZ-like states in the Qubit-Qudit Rabi model

Yuan Shen[1], Giampiero Marchegiani[2*], Gianluigi Catelani[3,2], Luigi Amico[2,4,5,6,7],
Ai Qun Liu[1†], Weijun Fan[1‡] and Leong-Chuan Kwek[1,4,7,8§]

**1** School of Electrical and Electronic Engineering, Nanyang Technological University,
Block S2.1, 50 Nanyang Avenue, Singapore 639798
**2** Quantum Research Centre, Technology Innovation Institute, Abu Dhabi, UAE
**3** JARA Institute for Quantum Information (PGI-11), Forschungszentrum Jülich,
52425 Jülich, Germany
**4** Centre for Quantum Technologies, National University of Singapore,
3 Science Drive 2, Singapore 117543
**5** INFN-Sezione di Catania, Via S. Sofia 64, 95127 Catania, Italy
**6** LANEF 'Chaire d'excellence', Université Grenoble-Alpes & CNRS, F-38000 Grenoble, France
**7** MajuLab, CNRS-UNS-NUS-NTU International Joint Research Unit, UMI 3654, Singapore
**8** National Institute of Education, Nanyang Technological University,
1 Nanyang Walk, Singapore 637616

⋆ giampiero.marchegiani@tii.ae † EAQLiu@ntu.edu.sg
‡ EWJFan@ntu.edu.sg §kwekleongchuan@nus.edu.sg

## Abstract

We study a Rabi type Hamiltonian system in which a qubit and a *d*-level quantum system (qudit) are coupled through a common resonator. In the weak and strong coupling limits the spectrum is analysed through suitable perturbative schemes. The analysis show that the presence of the multilevels of the qudit effectively enhance the qubit-qudit interaction. The ground state of the strongly coupled system is found to be of Greenberger-Horne-Zeilinger (GHZ) type. Therefore, despite the qubit-qudit strong coupling, the nature of the specific tripartite entanglement of the GHZ state suppresses the bipartite entanglement. We analyze the system dynamics under quenching and adiabatic switching of the qubit-resonator and qudit-resonator couplings. In the quench case, we found that the non-adiabatic generation of photons in the resonator is enhanced by the number of levels in the qudit. The adiabatic control represents a possible route for preparation of GHZ states. Our analysis provides relevant information for future studies on coherent state transfer in qubit-qudit systems.

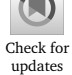

# 1 Introduction

The quantum Rabi model (QRM) describes the interaction between a two-level system and a single quantized harmonic oscillator mode. It is one of the most celebrated models in atomic physics for light-matter interaction [1]. In quantum technology, Rabi-like models are widely employed to describe the effective physics emerging in a variety of different contexts ranging from spintronics [2,3] to trapped ions [4], and from circuit quantum electrodynamics (cQED) [5] to atom-superconducting qubit hybrid schemes [6]. Despite its simple form, the Rabi model was solved exactly only recently [7,8]. The ground state of the quantum Rabi model consists of a non-classical highly entangled state of two-level system and bosonic mode [7,9]. In cQED, different regimes of interaction between the two-level system and the bosonic field can be explored. In particular, weak and strong coupling regimes are routinely exploited for read-out and coherent state transfer [10]. Recent studies have demonstrated the possibility of reaching the ultrastrong and deep strong coupling regimes, too [11,12].

Here, we formulate and study a Rabi-type minimal model describing qubit-qudit interaction mediated by a single mode quantum bosonic field. This type of models has emerged recently in several studies of specific systems where atoms, solid state devices (such as superconducting and quantum dot qubits) are assembled together to form hybrid quantum networks [6,13–20]. In this context, entangling quantum systems of heterogeneous nature is sought intensively [21–23], as such hybrid entangled states could become useful in converting quantum information between different encodings [24].

We shall see that the physics of our model is particularly interesting in the ultrastrong coupling regime [18,25–28]. In particular, the ground state of the system turns out to be defining a Greenberger-Horne-Zeilinger (GHZ) entangled state. GHZ states present great significance among all types of multipartite entanglement [29]. These states exhibit maximal correlations

between three or more quantum systems. GHZ states have been considered a key resource in fundamental physics since the early stages of quantum information. They have also been proven useful in various quantum technologies, including quantum error-correcting codes [30] and quantum metrology beyond the Heisenberg limit [31].

We point out that the generation of GHZ hybrid entanglement defines a challenging problem in quantum technology. In cQED, for example, GHZ hybrid entanglement has been achieved by state-dependent phase shift operations which involve complicated control and feedback sequences [23,32]. In this context, exploration of the ultrastrong coupling regime has been demonstrated beneficial for GHZ state preparation [33,34]. Indeed, our scheme guarantees a straightforward preparation of hybrid GHZ states, as such states could appear in the ground state of the QRM at large coupling strengths.

The article is organized as follows: in Sec. 2, we introduce a generalization of the quantum Rabi model to describe the qubit-qudit interaction through the resonator bosonic field. We demonstrate, through an analytical approach based on adiabatic approximation and perturbative expansion, that hybrid GHZ states constitute the ground state solutions in the ultrastrong coupling limit. In Sec. 3, we study the bipartite entanglement between the qubit and qudit mediated by the common resonator, quantified by negativity [35]. We show that the presence of the GHZ state induces an exponential suppression of the negativity for large values of the coupling strengths. Dynamical features are investigated in Sec. 4. In particular, we show the dynamics after quenching the coupling strength, and propose adiabatic state preparation schemes for the hybrid GHZ states. A short discussion of the main results of the manuscript is presented in Sec. 5.

## 2 The qubit-qudit Rabi model

We investigate the physical system schematically pictured in Fig. 1. The scheme features a qubit (two-level quantum system) and a qudit ($d$-level quantum system) individually coupled to a common quantum resonator described by bosonic degrees of freedom. The Hamiltonian reads (here, and in the rest of this article, we work in units $\hbar = 1$)

$$\hat{H} = \omega \hat{a}^\dagger \hat{a} - \frac{\Omega_1}{2} \hat{\sigma}_z + \Omega_2 \hat{J}_d^z + [g_1 \hat{\sigma}_x + g_2 (\hat{J}_d^+ + \hat{J}_d^-)](\hat{a}^\dagger + \hat{a}). \tag{1}$$

Here, $\hat{\sigma}_{x,y,z}$ are the Pauli matrices for the qubit with transition frequency $\Omega_1$, $\hat{J}_d^{z,\pm}$ are the spin $(d-1)/2$ operators with level spacing $\Omega_2$, and $\hat{a}(\hat{a}^\dagger)$ is the annihilation (creation) operator for the bosonic field with frequency $\omega$. The coupling strengths $g_{1,2}$ in Eq. (1) quantify the vacuum-Rabi splittings. Employing a jargon that is widely used in the literature, we will denote the qubit and the qudit as "artificial atoms". For $d = 2$, our model is equivalent (up to a sign convention) to the two-qubit quantum Rabi model [36–39]. In contrast to the single qubit Rabi model, this generalized model is not integrable for general parameter values [37,38].

The eigenvalues and eigenstates of $\hat{H}$ can be readily obtained numerically. Here, we devise analytical approximation schemes both in the weak-coupling and in the ultrastrong coupling regimes.

The weak coupling limit ($g_1, g_2 \ll \omega$), in the presence of strong qubit/qudit-resonator detuning ($\Omega_1, \Omega_2 \ll \omega$), can be treated by means of a Schrieffer-Wolff transformation [5,40]. In particular, we apply the following unitary transformation to the Hamiltonian Eq. (1):

$$\hat{V} = \exp(\hat{S}) = \exp[\epsilon_1 (\hat{a}^\dagger \hat{\sigma}_+ - \hat{a} \hat{\sigma}_-) + \xi_1 (\hat{a}^\dagger \hat{\sigma}_- - \hat{a} \hat{\sigma}_+)$$
$$+ \epsilon_2 (\hat{a}^\dagger \hat{J}_d^+ - \hat{a} \hat{J}_d^-) + \xi_2 (\hat{a}^\dagger \hat{J}_d^- - \hat{a} \hat{J}_d^+)], \tag{2}$$

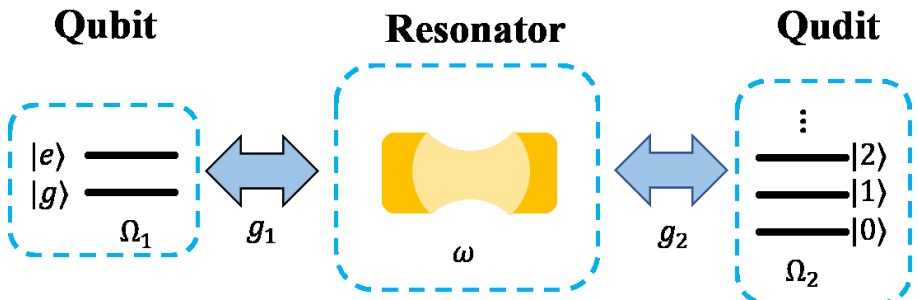

Figure 1: Model schematics. The system is composed of a qubit with level spacing $\Omega_1$, an harmonic oscillator (resonator) with characteristic frequency $\omega$, and a $d$-level quantum system (qudit) with level spacing $\Omega_2$. The qubit and the qudit are coupled to the resonator through the coupling constants $g_{1,2}$.

where we choose

$$\epsilon_i = \frac{g_i}{\omega - \Omega_i} = \frac{g_i}{\Delta_i}, \tag{3}$$

$$\xi_i = \frac{g_i}{\omega + \Omega_i} = \frac{g_i}{\Sigma_i}. \tag{4}$$

In the weak coupling limit considered here $\epsilon_i, \xi_i \ll 1$. In particular, at the lowest order in the expansion, the effective Hamiltonian $\hat{H}_{\text{eff}} = \hat{V}\hat{H}\hat{V}^\dagger$ reads:

$$\hat{H}_{\text{eff}} \simeq \hat{H}_0 - \frac{1}{2}\Big[[g_1\hat{\sigma}_x + g_2(\hat{J}_d^+ + \hat{J}_d^-)](\hat{a}^\dagger + \hat{a}), \hat{S}\Big]$$

$$= \tilde{\omega}\hat{a}^\dagger\hat{a} - \frac{\tilde{\Omega}_1}{2}\hat{\sigma}_z + \tilde{\Omega}_2\hat{J}_d^z - g_{\text{eff}}\hat{\sigma}_x(\hat{J}_d^+ + \hat{J}_d^-)$$

$$- \frac{1}{2}g_2(\epsilon_2 + \xi_2)\left[\frac{d^2-1}{2} - 2(\hat{J}_d^z)^2 + (\hat{J}_d^+)^2 + (\hat{J}_d^-)^2\right]$$

$$- \frac{1}{2}g_1(\epsilon_1 + \xi_1), \tag{5}$$

with renormalized frequencies:[1]

$$\tilde{\omega} = \omega + g_1(\epsilon_1 - \xi_1)\hat{\sigma}_z + 2g_2(\epsilon_2 - \xi_2)\hat{J}_d^z, \tag{6}$$

$$\tilde{\Omega}_1 = \Omega_1 - g_1(\epsilon_1 - \xi_1), \tag{7}$$

$$\tilde{\Omega}_2 = \Omega_2 - g_2(\epsilon_2 - \xi_2), \tag{8}$$

and effective coupling $g_{\text{eff}} = [g_1(\epsilon_2 + \xi_2) + g_2(\epsilon_1 + \xi_1)]/2$. In Eq. (5), $\hat{H}_0 = \omega\hat{a}^\dagger\hat{a} - \frac{\Omega_1}{2}\hat{\sigma}_z + \Omega_2\hat{J}_d^z$ is the uncoupled Hamiltonian and $[\dots,\dots]$ denotes the commutator. Within our approximation, the energy spectrum consists of different manifolds characterized by a fixed value of resonator photon number operator $\hat{N} = \hat{a}^\dagger\hat{a}$ (the interactions between different manifolds can be neglected due to the large resonator frequency compared to other energy scales). For the qubit ($d = 2$), we have $(\hat{J}_2^\pm)^2 = 0$, $(\hat{J}_2^z)^2 = 1/4$, and the spectrum of the Hamiltonian in Eq. (5) can be found by diagonalizing a $4 \times 4$ matrix consisting of two $2 \times 2$ blocks. In the lowest manifold ($N = 0$), the four eigenenergies are given by

$$E_{d=2} = \pm\sqrt{\frac{1}{4}(\tilde{\Omega}_1 \pm \tilde{\Omega}_2)^2 + g_{\text{eff}}^2} - \sum_{i=1,2}\frac{1}{2}g_i(\epsilon_i + \xi_i). \tag{9}$$

---

[1]Here our notations is slightly abusive, since $\tilde{\omega}$ contains the operators $\hat{\sigma}_z, \hat{J}_d^z$. However, since we focus on the $N = 0$ subspace of the resonator degree of freedom, this notation simplifies the subsequent discussion.

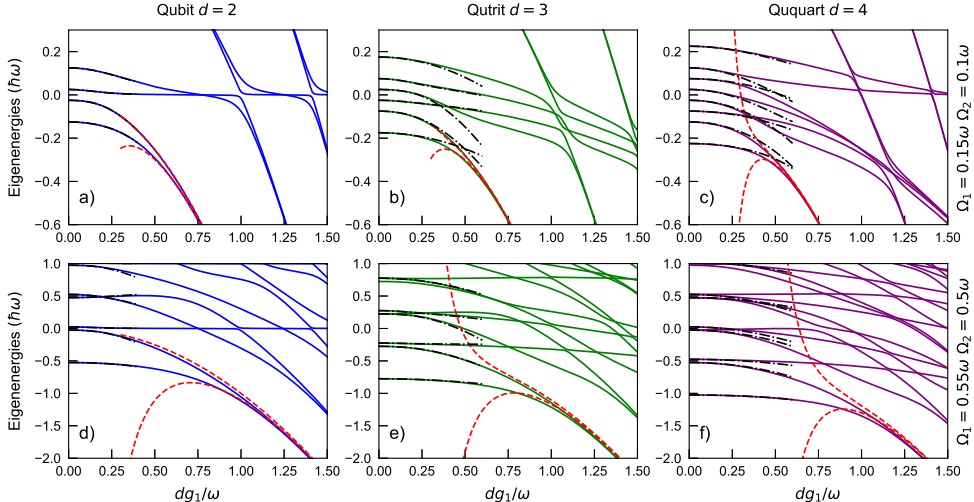

Figure 2: Hamiltonian energy spectrum with qudit dimensions $d = 2, 3, 4$. (a)-(f) Low energy spectrum vs coupling strength $g$ obtained through numerical diagonalization of the full Hamiltonian Eq. (1) (solid). The numerical results are compared to the low coupling approximations (black dotted-dashed) of Eq. (9) (see also Appendix A for the qutrit and the ququart cases), and $d$-order perturbation approximation Eq. (11) (see also Appendix C for the qutrit and the ququart cases) in the ultrastrong coupling regime (red dashed), for $g_1 = g_2 = g$ and (top panels) $\Omega_1 = 0.15\omega, \Omega_2 = 0.1\omega$, (bottom panels) $\Omega_1 = 0.55\omega, \Omega_2 = 0.5\omega$.

In the general qudit case ($d \geq 3$), the eigenenergies can be obtained by computing the roots of the product of two degree $d$ characteristic polynomials of submatrices of dimension $2 \times d$. Simplified expression can be obtained by neglecting the $(\hat{J}_d^{\pm})^2$ terms in Eq. (5), and performing an approximation similar to the standard rotating wave approximation $\hat{\sigma}_x(\hat{J}_d^+ + \hat{J}_d^-) \sim \hat{\sigma}_+\hat{J}_d^- + \hat{\sigma}_-\hat{J}_d^+$, valid for $\Omega_1 \sim \Omega_2$. The approximate results in the $N = 0$ subspace for the qutrit and the ququart are reported in Appendix A.

In the ultrastrong coupling regime, the numerical results to be presented below are corroborated by an analytical approach combining the adiabatic approximation in the displaced oscillator basis [41] and degenerate perturbation theory. More precisely, we first obtain the exact spectrum of the reduced Hamiltonian

$$\tilde{H} = \omega\hat{a}^{\dagger}\hat{a} + g_1\hat{\sigma}_x(\hat{a}^{\dagger} + \hat{a}) + g_2(\hat{J}_d^+ + \hat{J}_d^-)(\hat{a}^{\dagger} + \hat{a}), \tag{10}$$

neglecting the free Hamiltonian of the qubit and qudit in the limit where $\Omega_1, \Omega_2 \ll \omega$, and $g_1, g_2 \lesssim \omega$. Then, these terms are restored within a perturbative expansion. The eigenstates of $\tilde{H}$ are product states $|\sigma\, m\, N_{\sigma,m}\rangle = |\sigma\rangle \otimes |m\rangle \otimes |N_{\sigma,m}\rangle$. Here, $\sigma = \uparrow, \downarrow$ are the eigenstates of $\hat{\sigma}_x$ with eigenvalues $\pm 1$, $|m = 0, 1, \dots, d-1\rangle$ are the eigenstates of the qudit operator $(\hat{J}_d^+ + \hat{J}_d^-)$ with eigenvalues $\lambda_m = -(d-1) + 2m$, and $|N_{\sigma,m}\rangle$ are displaced Fock states [36, 37, 41] (see Appendix B). The system yields a two-fold degenerate ground state, obtained from a displacement of the vacuum state in the resonator $\{|\uparrow, +, 0_{\uparrow,+}\rangle, |\downarrow, -, 0_{\downarrow,-}\rangle\}$, with energy $E_0 = -[g_1 + (d-1)g_2]^2/\omega$, where $|+\rangle$ ($|-\rangle$) is the eigenstate of the operator $(\hat{J}_d^+ + \hat{J}_d^-)$ with the largest(smallest) eigenvalue, i.e., $d-1(-d+1)$. The corrections to the spectrum of $\tilde{H}$ are then evaluated through perturbation theory in $\hat{H}' = -\frac{1}{2}\Omega_1\hat{\sigma}_z + \Omega_2\hat{J}_d^z$. The lowest (second) order corrections to the energy are obtained as (see Appendix B and Appendix C for a detailed

derivation):

$$\mathcal{E}_\pm = E_0 - \frac{\omega}{16(d-1)g_1 g_2} \Omega_1^2 e^{-4g_1^2/\omega^2}$$

$$- \omega \frac{(d-1)}{16(d-2)g_2^2 + 16 g_1 g_2} \Omega_2^2 e^{-4g_2^2/\omega^2}$$

$$\pm \delta_{d,2} \frac{\omega \Omega_1 \Omega_2}{8(d-1)g_1 g_2} e^{-2[g_1^2 + (d-1)^2 g_2^2]/\omega^2} , \qquad (11)$$

where $\delta_{i,j}$ is the Kronecker delta.[2] Notably, the two-fold degeneration of ground state is only resolved at $d$-order perturbation theory in $\hat{H}'$, with a correction proportional to $\Omega_1 \Omega_2^{d-1}$ (see Appendix C). The corresponding eigenstates read

$$|\Psi_\pm\rangle = \frac{1}{\mathcal{C}}[|\uparrow,+,0_{\uparrow,+}\rangle \pm |\downarrow,-,0_{\downarrow,-}\rangle + \sum_{(\sigma,m)\neq(\uparrow,+),(\downarrow,-)} C_{\sigma,m}(g_1,g_2)|\sigma,m,0_{\sigma,m}\rangle], \qquad (12)$$

where $\mathcal{C}$ is the normalization factor, and the functions $C_{\sigma,m}(g_1,g_2) \propto e^{-g_1^2/\omega^2}, e^{-g_2^2/\omega^2}$ are exponentially suppressed with the coupling strengths. As discussed above, the analytical expression in Eqs. (11)-(12) are expected to hold in the regime where the free Hamiltonian terms of the atoms are treated as perturbations to the interacting system, i.e. $\Omega_1, \Omega_2 \ll \omega, g_1, g_2$.

In Fig. 2, we display the low-energy spectrum of the Hamiltonian in Eq. (1) as a function of the coupling strength (with $g_1 = g_2$) for $d = 2$ (panels a,d), $d = 3$ (panels b,e), and $d = 4$ (panels c,f). For visualization purposes, we plot the energy as a function of $dg_1$ (with $d$ number of levels in the qudit), since the ground state energy for $g_1 = g_2$ scales as $E_0/\omega \sim -(dg_1/\omega)^2$ for large values of $g_1$. The analytical expressions obtained in the low-coupling limit (dotted-dashed) and through perturbation theory in the ultrastrong coupling regime (dashed) are compared with the solutions obtained through numerical diagonalization (solid). In the low coupling regime, the analytical expression of Eq. (9) gives a very accurate description of the spectrum for $dg_1 \lesssim 0.4\omega$ (see Fig. 2a), and Fig. 2d)). For the general qudit case, the expressions obtained through RWA approximation reproduce the numerical results in a less satisfactory way. Still, they correctly reproduce the spectrum for $dg_1 \lesssim 0.3\omega$.

For $\Omega_1 = 0.15\omega, \Omega_2 = 0.1\omega$ (top panels), an excellent agreement between the strong coupling regime approximations and numerical solutions arises for $dg_1, dg_2 \gtrsim 0.3\omega$. For higher $\Omega_1$ and $\Omega_2$ values ($\Omega_1 = 0.55\omega, \Omega_2 = 0.5\omega$, bottom panels), the adiabatic approximation breaks down below $dg_1, dg_2 \sim 0.75\omega$ (Fig. 2b).

With increasing coupling strengths $g_1, g_2$, the higher order correction terms in Eq. (12) are suppressed exponentially, and the states $|\Psi_\pm\rangle$ approach the GHZ-type states. Note that the states $|0_{\uparrow,+}\rangle = D^\dagger(\frac{g_1+(d-1)g_2}{\omega})|0\rangle$ and $|0_{\downarrow,-}\rangle = D^\dagger(-\frac{g_1+(d-1)g_2}{\omega})|0\rangle$ are coherent states with opposite displacement in the phase space, which are asymptotically orthogonal in the limit $g_1, g_2 \to \infty$. As a result, the ground state under such large coupling assumption can be approximated as:

$$|\Psi_{GHZ}\rangle = \frac{1}{\sqrt{2}}(|\uparrow,+,0_{\uparrow,+}\rangle \pm |\downarrow,-,0_{\downarrow,-}\rangle). \qquad (13)$$

The validity of this approximation is investigated in Fig. 3, where the fidelity between the GHZ state and the ground state of the Hamiltonian obtained through numerical diagonalization is plotted as a function of the coupling strength $g_1 = g_2$. In this manuscript, we define the fidelity between two pure states $|\phi\rangle, |\psi\rangle$ as $\mathcal{F} = |\langle\phi|\psi\rangle|$. In agreement with our perturbative calculation, the fidelity of the ground state with the GHZ state approaches the unit value in the limit $g_1 \gg \Omega_1, \Omega_2$.

---

[2]We have verified that for $d = 2$, when both equations (9) and (11) are applicable (that is, when taking $g_i, \Omega_i \to 0$) they agree at next to leading order.

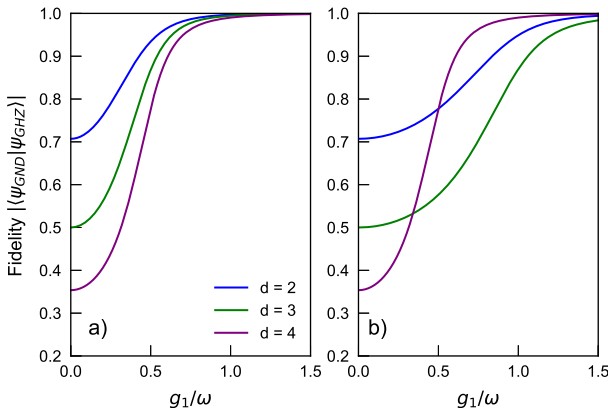

Figure 3: Ground state vs GHZ state for $g_1 = g_2$ and different qudit sizes $d = 2, 3, 4$ (top to bottom at $g_1 = 0$). Fidelity between the Hamiltonian ground state (obtained through numerical diagonalization) and the GHZ state Eq. (13) as a function of the coupling strength for (a) $\Omega_1 = 0.15\omega, \Omega_2 = 0.1\omega$, (b) $\Omega_1 = 0.55\omega, \Omega_2 = 0.5\omega$.

# 3   Negativity

In our scheme, it is interesting to investigate the bipartite entanglement between the qubit and qudit. We choose to compute negativity [35] as the measure of entanglement. For a bipartite pure state $|\varphi\rangle_{AB}$ in a $d \otimes d'(d \leq d')$ quantum system, the negativity is defined as

$$\mathcal{N}_{|\varphi\rangle_{AB}} = \frac{1}{2}(\||\varphi\rangle_{AB}\langle\varphi|^{T_B}\|_1 - 1), \tag{14}$$

where $|\varphi\rangle_{AB}\langle\varphi|^{T_B}$ is the partial transpose of $|\varphi\rangle_{AB}\langle\varphi|$ and $\|\cdot\|_1$ is the trace norm. To extract bipartite pair-wise entanglement in a tripartite system, we use the reduced density matrix of $|\varphi\rangle_{ABC}$ on subsystem $A \otimes B$ by tracing over subsystem $C$: $\rho_{AB} = \mathbf{tr}_C|\varphi\rangle_{ABC}\langle\varphi|$.

Figure 4 displays the density plot of the negativity as a function of the coupling strengths $g_1, g_2$ in the qubit (Fig. 4a), qutrit (Fig. 4b), ququart (Fig. 4c) cases, for $\Omega_1 = \Omega_2 = 0.1\omega$. Note

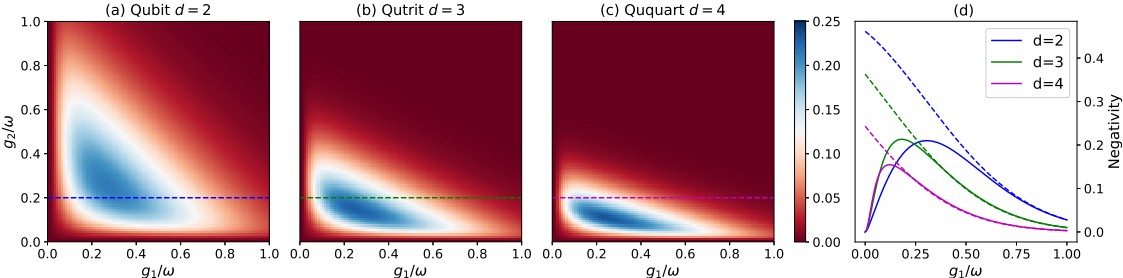

Figure 4: Coupling dependence of the ground state negativity between the qubit and the qudit. (a)-(c) Density plots of the ground state negativity as a function of $g_1$ and $g_2$ for the (a) qubit-qubit, (b) qubit-qutrit, (c) qubit-ququart cases. The plots are obtained by numerical calculations for $\Omega_1 = \Omega_2 = 0.1\omega$. (d) Cuts of the density plots for $g_2 = 0.2\omega$, indicated by the dashed lines in panels (a)-(c). The results obtained through numerical calculations (solid) are compared with the approximate expression of the negativity Eq. (16) (dashed), obtained in the ultrastrong coupling regime.

that the negativity is clearly symmetric under the exchange $g_1 \leftrightarrow g_2$ in the qubit case (since $\Omega_1 = \Omega_2$), and it becomes gradually more asymmetric by increasing the number of levels. The bipartite entanglement between the two artificial atoms has a nontrivial response to the coupling strengths between subsystems. In particular the negativity is maximum for intermediate values of the couplings, and it is strongly suppressed at large couplings. The parameters where the maximum negativity is obtained depend on the number of levels in the qudit and read: $g_1 = g_2 \simeq 0.24\omega$ (qubit), $g_1 \simeq 0.21\omega$, $g_2 \simeq 0.17\omega$ (qutrit), $g_1 \simeq 0.21\omega$, $g_2 \simeq 0.14\omega$ (ququart).

For a better visualization, in Fig. 4d we consider cuts of Figs. 4a-c at a fixed value of $g_2 = 0.2\omega$. The negativity first rises to a maximum with increased coupling strength $g_1$, before decaying to zero exponentially (as we will discuss below). This phenomenon is reported in Ref. [42] where an approximate expression is derived to explain the curve at weaker coupling. Here, we obtain an analytical expression for the decaying curve. In addition, our approach demonstrates that the entanglement suppression is a consequence of the structure of the entanglement encoded in the ground state [see Eq. (13)]: the tripartite GHZ state at large coupling limit($g_1, g_2 \to \infty$), hinders the bipartite entanglement obtained after tracing out one of the subsystems, that asymptotically vanishes. This property of the GHZ states results in a counter-intuitive implication: the strong coupling can destroy entanglement between the two quantum systems connected by the resonator.

Now we show that the approximate expression of the ground state, i.e., the GHZ state of Eq. (13), leads to an accurate prediction for the entanglement at large couplings; we can easily calculate the negativity for those states. Indeed, the corresponding reduced density matrix $\rho'$ with resonator degree-of-freedom traced out is a $(2d \times 2d)$ matrix and has only four non-zero matrix elements:

$$\rho' = \frac{1}{2} \begin{bmatrix} 1 & \dots & K \\ \vdots & \ddots & \vdots \\ K & \dots & 1 \end{bmatrix}, \tag{15}$$

with $K = \exp\{-2[g_1 + (d-1)g_2]^2/\omega^2\}$. Therefore the analytical expression for the negativity of the ground state in Eq. (13) is

$$\mathcal{N} = \frac{1}{2}|K| = \frac{1}{2}\exp\{-2[g_1 + (d-1)g_2]^2/\omega^2\}. \tag{16}$$

These approximate expressions are displayed (dashed) in Fig. 4. Note that the approximation describes very accurately the exponential decay of the negativity at large couplings.

We close the section by noting that the negativity measure is not a sufficient test of entanglement for systems with dimensions beyond $2 \times 3$. Under such circumstances, a state with zero negativity could possibly be a positive partial transpose (PPT) or "bound entangled" state, which is argued to be metrologically useful [43–45].

# 4 Dynamics

In this section, we discuss the dynamical evolution of the coupled system. We consider two complementary cases: the quench dynamics starting from the non interacting state and the adiabatic preparation of the GHZ state.

## 4.1 Quench dynamics

We start by discussing the dynamics of the system under non-adiabatic switching of the interaction. We consider the system initially prepared in the ground state of the uncoupled

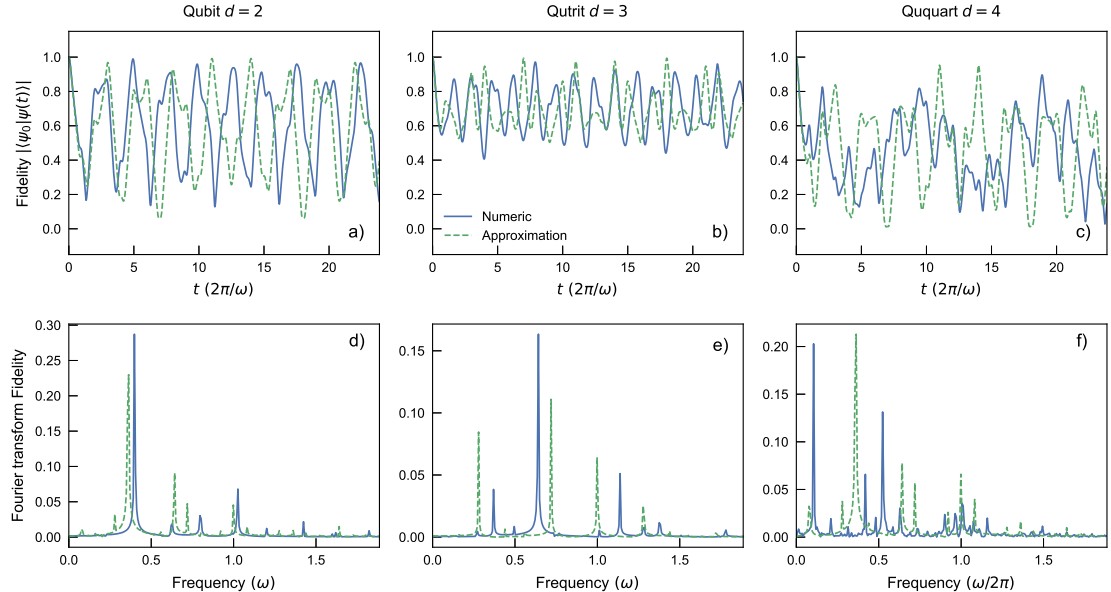

Figure 5: Fidelity dynamics after quenching the interaction strength. a)-b)-c) Fidelity between the initial state of the system and the instantaneous state for qubit (a), qutrit (b) and ququart (c) cases. The initial state is the ground state of the uncoupled system. The numerical expressions (solid) are compared to approximations (dashed) derived in the main text Eq. (18). d)-e) -f) Fast fourier transform of the fidelity evolution for d) qubit, e) qutrit, f) ququart. Parameters are $\Omega_1 = 0.12\omega, \Omega_2 = 0.1\omega$, and $g_1 = g_2 = 0.3\omega$.

Hamiltonian $\hat{H}_0$, *i.e.* $|\psi_0\rangle = |g\,0\,0\rangle$. For simplicity, we consider an instantaneous quench of the coupling constants to the final values $g_1 = g_2 = 0.3\omega$. In this case, the time-evolved state reads

$$|\psi(t)\rangle = e^{-i\hat{H}t}|\psi_0\rangle. \tag{17}$$

Due to the non-adiabatic control, the state of the system is different from the ground state of the interacting Hamiltonian after the quench, and evolves in time.

Figures 5a-c display the time evolution of the fidelity between the initial state $|\psi_0\rangle$ and the time evolved state $|\psi(t)\rangle = e^{-i\hat{H}t}|\psi_0\rangle$ in the qubit, qutrit, and ququart cases, respectively, for $\Omega_1 = 0.12\omega$ and $\Omega_2 = 0.1\omega$. We note that the dynamics of $|\psi_0\rangle$ involves multiple frequencies, and displays a clear dependence on the number of levels of the qudit. An approximate description of the evolution can be obtained by neglecting the free terms of the atoms in the Hamiltonian. Namely, we set $\Omega_{1,2} = 0$, and expand $|\psi_0\rangle$ in the eigenstates basis of $\tilde{H}$ [see Eq. (10)], *i.e.*, $|\psi_0\rangle = \sum_{\sigma mN} \langle \sigma\, m\, N_{\sigma m}|g\,0\,0\rangle |\sigma\, m\, N_{\sigma m}\rangle$. For the qubit case, we obtain

$$\mathcal{F}^{d=2} = \frac{1}{2}\left| \sum_{N=0}^{+\infty} e^{-iE_{\uparrow+}t}|\langle N_{\uparrow,+}|0\rangle|^2 + e^{-iE_{\uparrow-}t}|\langle N_{\uparrow,-}|0\rangle|^2 \right|, \tag{18}$$

where $E_{\uparrow\pm} = \omega(N - \alpha_\pm^2)$ and $|\langle N_{\uparrow,\pm}|0\rangle|^2 = e^{-\alpha_\pm^2}\alpha_\pm^{2N}/N!$, with $\alpha_\pm = (g_1 \pm g_2)/\omega$. For the qutrit and the ququart case the summation in Eq. (18) involves $d$ terms. Such approximated dynamics is displayed[3] in Figs. 5a-c using dashed curves. We note that the approximate expressions capture the initial decrease in fidelity and the amplitude of its oscillations, as well

---

[3]In the infinite sum of Eq. (18), we retained terms up to $N_{max} = 10$, since we verified that the convergence is quite rapid for our parameter values.

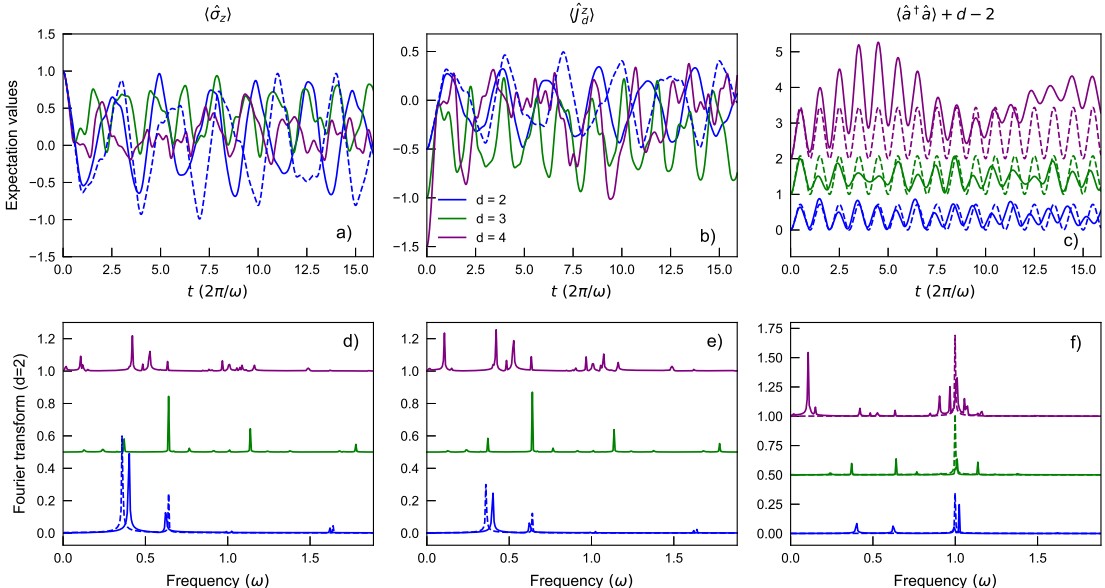

Figure 6: Operator dynamics after quenching the interaction strength. a)-b)-c) Dynamics of the expectation value of a) the qubit population, b) qudit population and c) mean photon number. The dashed lines in panel a)-c) are given by Eqs. (19),(20). d)-e)-f) Fast Fourier transform of the expectation value evolution for d) qubit population, e) qudit population, f) photon number population. The curves are vertically offset by $(d-2)/2$ for visualization purposes. Parameters are $\Omega_1 = 0.12\omega$, $\Omega_2 = 0.1\omega$, and $g_1 = g_2 = 0.3\omega$.

as the main frequency components of the evolution. We carried out the frequency analysis of the dynamical response through the fast Fourier transform of the curves displayed in Figs. 5a-c.[4] The results are reported in Figs. 5d-f. In the qubit case ($d = 2$, Fig. 5d), the analytical expression of Eq. (18) approximately captures the dominant frequency of the oscillation, with a small shift towards smaller frequencies. For the qutrit ($d = 3$, Fig. 5e) and the ququart ($d = 4$, Fig. 5f), the analytical expression gives a good estimation of the number of harmonics in the time oscillations; the actual values of the frequencies, instead, are obtained with less precision. The discrepancies arise because the approximated analytical scheme neglects the free Hamiltonians of the qubit and the qudit [the second and third terms in the right hand side of Eq. (1)].

Figure 6 displays the time evolution of the expectation values of $\hat{\sigma}_z$ (panel a), $\hat{J}_d^z$ (panel b), and the mean photon number $\hat{a}^\dagger \hat{a}$ (panel c) for the qubit, qutrit and ququart cases. Consider first the qubit population $\langle \hat{\sigma}_z(t) \rangle$: in all the three cases, the evolution is non-harmonic, and the number of relevant frequencies increases by increasing the number of levels in the qudit. In particular, for precise modeling of the dynamic in this regime different Fock number states must be included in the calculation and neglecting the off-diagonal terms in the basis of $\tilde{H}$ would provide a poor description of the physics (see the above discussion).

Consider for instance, the time evolution of $\hat{\sigma}_z$ in the qubit case (blue curve in Fig. 6a). We can work within the approximation exploited above for the strong coupling regime. Namely, we approximate $\hat{H}$ with $\tilde{H}$ of Eq. (10), and we write $|\psi_0\rangle$ in the basis of $\tilde{H}$. By performing the calculation, it is possible to derive a simple expression for $\langle \hat{\sigma}_z \rangle$ when $g_1 = g_2$ (as in the plot

---

[4]For better visualization, we subtract the mean value from each curve before performing the Fourier transform, hence removing the zero-frequency peaks.

of Fig. 6a), namely

$$\langle \hat{\sigma}_z(t) \rangle \sim \sum_{N=0}^{+\infty} \frac{(4g_1^2)^N}{N!} \cos(4g_1^2 t/\omega - N\omega t). \tag{19}$$

This approximate expression is displayed in Fig. 6a (dashed). In agreement with the results displayed for the fidelity in Fig. 5a, the approximation leading to Eq. (19) gives good estimates for both the amplitude and the main frequency of the oscillations. This is confirmed by the Fourier analysis in Fig. 6d. Similar considerations apply to the time evolution of the expectation value of $\langle \hat{J}_d^z \rangle$. Since the qubit system is symmetric under the exchange $1 \leftrightarrow 2$, the approximated evolution $\hat{H} \sim \tilde{H}$ can be readily obtained from Eq. (19); recalling that $\hat{J}_{d=2}^z = \hat{\sigma}_z/2$, we obtain $\langle \hat{J}_{d=2}^z(t) \rangle \sim -1/2 \sum_{N=0}^{+\infty} \frac{(4g_1^2)^N}{N!} \cos(4g_1^2 t/\omega - N\omega t)$ [the minus sign comes from our sign convention in Eq. (1)]. As a consequence, the Fourier transform of $\langle \hat{J}_{d=2}^z(t) \rangle$, shown in Fig. 6e, perfectly reproduces the one displayed in Fig. 6d, up to a factor 2.

Notably, the dynamics of the mean photon number $\langle \hat{a}^\dagger \hat{a} \rangle$ is much more regular, as displayed in Fig. 6c; for visualization purposes, the various curves are offset by the quantity $d - 2$. The plot clearly displays two relevant features: i) there is a frequency mode which is independent of the number of levels in the qudit; ii) the amplitude of the oscillations increases with $d$. These features can be discussed retaining the approximation $\hat{H} \sim \tilde{H}$. In particular, by applying the Baker-Hausdorff expansion to the operator $e^{i\tilde{H}t} \hat{a}^\dagger \hat{a} e^{-i\tilde{H}t}$, we derive

$$\langle \psi(t) | \hat{a}^\dagger \hat{a} | \psi(t) \rangle = 4[g_1^2 + (d-1)g_2^2] \sin^2(\omega t/2). \tag{20}$$

The validity of this approximation is investigated in Fig. 6c (dashed curves). While we find a good agreement for the qubit and the qutrit cases, the deviations are more relevant for the ququart. This result is confirmed by the Fourier transform of the curves, displayed in Fig. 6f; note that the curves have a vertical offset of $(d-2)/2$ for visualization purposes. As discussed above, all the curves share a harmonic component with frequency $\omega$, captured by the adiabatic approximation Eq. (20). This mode represents the primary frequency component in the qubit ($d = 2$, blue) and qutrit cases ($d = 3$, green). The situation is more complex for the ququart ($d = 4$, purple), with several sidebands and an enhanced low-frequency oscillation $\sim 0.1\omega$.

We conclude this section by emphasizing that the presence of additional levels in the qudit is beneficial to inducing non-adiabatic photon generation in the ultrastrong coupling regime.

## 4.2 Adiabatic state preparation

In this section we demonstrate how the state in Eq. (13) can be prepared with high fidelity by adiabatic evolution:

$$\hat{H}(t) = [1 - \mu(t)]\hat{H}_{\text{in}} + \mu(t)\hat{H}. \tag{21}$$

The initial state and the final state correspond to the ground state of the Hamiltonians $\hat{H}_{\text{in}} = \hat{H}(t = 0)$ and $\hat{H}$, respectively, and $\mu(t)$ is a function that goes from 0 to 1 when $t$ goes from 0 to the final evolution time $t_f$; $\mu(t)$ is chosen to be a linear function in our simulations. Here we propose two simple schemes to prepare the hybrid GHZ state: I. switch on the couplings at fixed frequencies; II. change the frequencies at fixed couplings. In both schemes, the final Hamiltonian $\hat{H}$ is set to be the Hamiltonian of the qubit-resonator-qudit system in Eq. (1).

I. In the first approach, the system is initialized without coupling terms $g_1(t = 0) = g_2(t = 0) = 0$, i.e., $\hat{H}_{\text{in}} = \hat{H}_0 = \omega \hat{a}^\dagger \hat{a} - \frac{\Omega_1}{2}\hat{\sigma}_z + \Omega_2 \hat{J}_d^z$. During the adiabatic evolution, the coupling terms are gradually switched on to the final value $g_f = 0.5\omega$, reading $g_1(t) = g_2(t) = \mu(t)g_f$, see the inset in Fig. 7a. The main panel of Fig. 7a displays the evolution of the fidelity between the instantaneous state of the system under the time-dependent Hamiltonian $\hat{H}(t)$ and the expected GHZ state at the final time, obtained by setting

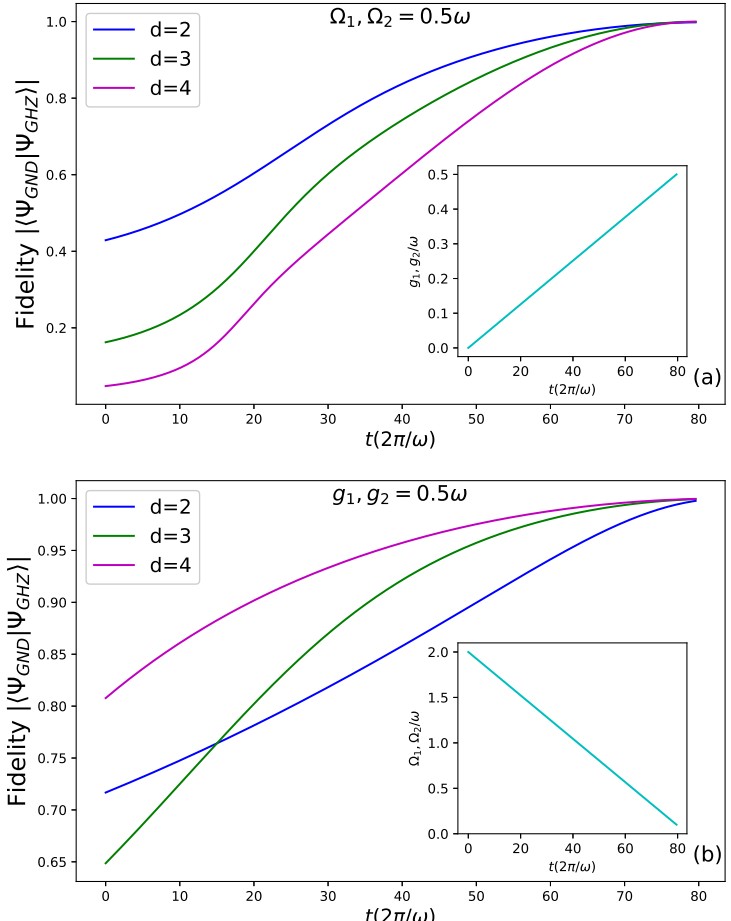

Figure 7: Adiabatic state preparation of the GHZ state in the qubit, qutrit, and ququart cases. (a) Time evolution of the fidelity between the instantaneous state and the hybrid GHZ state in Eq.(13). The coupling terms $g_1, g_2$ are adiabatically switched on at $t = 0$ and linearly increased to $0.5\omega$, as shown in the inset. (b) State fidelity with the GHZ state vs time in the adiabatic process where the atoms frequencies are linearly reduced from the initial value $\Omega_1, \Omega_2 = 2\omega$ to $0.1\omega$, as displayed in the inset.

$g_1 = g_2 = 0.5\omega$ in Eq. (13). The different curves corresponds to the qubit, qutrit and ququart cases. In all the cases, the fidelity is minimum at $t = 0$ and grows monotonically with time, reaching the unit value (within numerical accuracy) for $t = t_f = 500/\omega$. Note that for a given time the fidelity is maximum in the qubit case, and typically decreases by increasing the number of levels in the qudit.

II. Here, we keep fixed the coupling terms $g_1, g_2$, while tuning the characteristic frequencies of the artificial atoms. Specifically, the system is initialized to $\hat{H}_0 = \omega\hat{a}^\dagger\hat{a} - \Omega'_1\hat{\sigma}_z/2 + \Omega'_2\hat{J}^z_d + g_1\hat{\sigma}_x(\hat{a}^\dagger + \hat{a}) + g_2(\hat{J}^+_d + \hat{J}^-_d)(\hat{a}^\dagger + \hat{a})$, with the initial transition frequencies satisfying $\Omega'_1 \gg \Omega_1$ and $\Omega'_2 \gg \Omega_2$. In the adiabatic preparation, the artificial atoms frequencies $\Omega_1(t), \Omega_2(t)$, are linearly reduced to the final values $\Omega_1 = \Omega_2 = 0.1\omega$ (see the inset of Fig. 7b). The corresponding evolution of the fidelity with the final GHZ state is shown in Fig. 7b. As in the previous case, the fidelity grows monotonically to 1 as time approaches $t_f$ in all the cases. Notably, the presence of additional levels in the qudit is displayed to be beneficial to the process: for a given $t > t_f/2$, the fidelity is larger in the qutrit and ququart case with respect to the qubit case.

# 5 Conclusion

We formulated and studied a quantum Rabi type model describing the interaction between a two-level and a multi-level system mediated by a single mode bosonic field (in quantum technology such bosonic field is realized by a resonator). In the weak and in the strong coupling limits, we devised two different analytical schemes. In the weak-coupling limit, the effective Hamiltonian is obtained through a suitable Schrieffer-Wolff transformation. Assuming strong qubit/qudit-resonator detuning, the spectrum of the effective Hamiltonian has been obtained exactly. In the strong coupling limit, the ground state of the system is provided by a tripartite-entangled state of the GHZ type. A known feature of the GHZ states is that they do not allow bipartite entanglement between the three partners. As a result, qubit and qudit cannot be highly entangled, and the correlation between them drops exponentially with increasing coupling in the ultrastrong coupling regime. Such analysis is supported by the study of the negativity providing the sufficient condition for the qubit-qudit entanglement.

We analyzed the system dynamics both under quenching and adiabatic control of system parameters (either the couplings or the atoms' frequencies). The non-adiabatic nature of the quantum quench leads to the generations of photons in the resonator, with a magnified effect by increasing the number of levels in the qudit. By adiabatic control, the GHZ states can be prepared with high fidelity. Both the analysis of the spectrum and the dynamics indicate that the interaction is effectively magnified by the number of the levels of the qudit. The study of the dynamics gives preliminary information on the qubit-qudit coherent state transfer. Our work provides relevant information for applications in quantum technology, particularly for hybrid quantum networks design [21,22,24,46]. The proposed scheme for the GHZ preparation may be implemented in cQED platforms, where both ultrastrong and deep strong coupling regimes have been studied theoretically [47–49] and implemented experimentally [11, 12, 50–52]. Very recently, the ultrastrong coupling regime has been reached in hybrid semiconducting-superconducting technology [53], too.

The generalization of the Rabi model to the qudit system can be applied to discuss superconducting circuits based on transmon qubits. Indeed, the multi-level nature of the energy spectrum cannot be ignored in these elements due to the weak anharmonicity. These circuits have been extensively investigated, even in combination with semiconducting qubits [54]. On the theoretical side, our scheme can be investigated in a more general setting. A certainly interesting direction to go would be studying the system dynamics in the presence of dissipation [55].

# Acknowledgements

We thank Yvonne Gao, Dariel Mok, and Andrea Iorio for fruitful discussions. LCK is supported by the Ministry of Education and the National Research Foundation of Singapore. WJ Fan would like to acknowledge the support from NRF-CRP19-2017-01.

# A Low coupling spectrum

Here we report the low-coupling approximation of the effective Hamiltonian Eq. (5), obtained by neglecting the $(\hat{J}_d^{\pm})^2$ terms, and performing a RWA-like approximation. For the qutrit $d = 3$,

$$E_{0,1} = -\frac{1}{2}g_1(\epsilon_1 + \xi_1) - g_2(\epsilon_2 + \xi_2) \pm \left(\frac{\tilde{\Omega}_1}{2} - \tilde{\Omega}_2\right), \tag{A.1}$$

$$E_{2,3} = -\frac{1}{2}g_1(\epsilon_1 + \xi_1) - \frac{3}{2}g_2(\epsilon_2 + \xi_2) - \frac{1}{2}\tilde{\Omega}_2 \pm \frac{1}{2}\sqrt{\left[\tilde{\Omega}_1 + \tilde{\Omega}_2 + g_2(\epsilon_2 + \xi_2)\right]^2 + 8g_{\text{eff}}^2}, \tag{A.2}$$

$$E_{4,5} = -\frac{1}{2}g_1(\epsilon_1 + \xi_1) - \frac{3}{2}g_2(\epsilon_2 + \xi_2) + \frac{1}{2}\tilde{\Omega}_2 \pm \frac{1}{2}\sqrt{\left[\tilde{\Omega}_1 + \tilde{\Omega}_2 - g_2(\epsilon_2 + \xi_2)\right]^2 + 8g_{\text{eff}}^2}. \tag{A.3}$$

For the ququart $d = 4$,

$$E_{0,1} = -\frac{1}{2}g_1(\epsilon_1 + \xi_1) - \frac{3}{2}g_2(\epsilon_2 + \xi_2) \pm 2\left(\tilde{\Omega}_1 - 3\tilde{\Omega}_2\right),$$

$$E_{2,3} = -\frac{1}{2}g_1(\epsilon_1 + \xi_1) - \frac{7}{2}g_2(\epsilon_2 + \xi_2) \pm \frac{1}{2}\sqrt{\left(\tilde{\Omega}_1 + \tilde{\Omega}_2\right)^2 + 16g_{\text{eff}}^2}, \tag{A.4}$$

$$E_{4,5} = -\frac{1}{2}g_1(\epsilon_1 + \xi_1) - \frac{5}{2}g_2(\epsilon_2 + \xi_2) - \tilde{\Omega}_2 \pm \frac{1}{2}\sqrt{\left[\tilde{\Omega}_1 + \tilde{\Omega}_2 - 2g_2(\epsilon_2 + \xi_2)\right]^2 + 12g_{\text{eff}}^2},$$

$$E_{6,7} = -\frac{1}{2}g_1(\epsilon_1 + \xi_1) - \frac{5}{2}g_2(\epsilon_2 + \xi_2) + \tilde{\Omega}_2 \pm \frac{1}{2}\sqrt{\left[\tilde{\Omega}_1 + \tilde{\Omega}_2 + 2g_2(\epsilon_2 + \xi_2)\right]^2 + 12g_{\text{eff}}^2}. \tag{A.5}$$

# B Adiabatic approximation

To diagonalize Eq. (1), we generalize the adiabatic approximation, which was first adopted in [41] to find solution to the single spin quantum Rabi model. In the adiabatic limit $\Omega_1, \Omega_2 \ll \omega$, the Hamiltonian is approximated as:

$$\tilde{H} = \omega\hat{a}^\dagger\hat{a} + g_1\hat{\sigma}_x(\hat{a}^\dagger + \hat{a}) + g_2(\hat{J}_d^+ + \hat{J}_d^-)(\hat{a}^\dagger + \hat{a}), \tag{B.1}$$

where the free energy terms of the atoms have been dropped from Eq. (1).

We consider eigenstates of $\tilde{H}$ in the form $|\sigma m N_{\sigma,m}\rangle = |\sigma\rangle \otimes |m\rangle \otimes |N_{\sigma,m}\rangle$ where $\sigma = \uparrow, \downarrow$ are the eigenstates of $\hat{\sigma}_x$ with $\hat{\sigma}_x|\uparrow, \downarrow\rangle = \pm|\uparrow, \downarrow\rangle$, $|m\rangle$ are the eigenstates of the qudit spin operator $(\hat{J}_d^+ + \hat{J}_d^-)$. Next we find the eigenvalues of $\tilde{H}$:

$$\tilde{H}|\sigma m N_{\sigma,m}\rangle = E_{\sigma m}^N |\sigma m N_{\sigma,m}\rangle. \tag{B.2}$$

## B.1 Qubit case

The eigenstates of $(\hat{J}_2^+ + \hat{J}_2^-)$ in the qubit basis $|0, 1\rangle$ reads $|\pm\rangle = \frac{1}{\sqrt{2}}(1, \pm 1)^T$ with eigenvalues $\pm 1$. We can derive a set of four eigenvalue equations for the resonator eigenstates $|N_{\sigma,m}\rangle$:

$$\left[\hat{a}^\dagger\hat{a} + \frac{(g_1 \pm g_2)}{\omega}(\hat{a}^\dagger + \hat{a})\right]|N_{\uparrow,\pm}\rangle = \frac{E_{\uparrow,\pm}^N}{\omega}|N_{\uparrow,\pm}\rangle, \tag{B.3}$$

$$\left[\hat{a}^\dagger\hat{a} - \frac{(g_1 \pm g_2)}{\omega}(\hat{a}^\dagger + \hat{a})\right]|N_{\downarrow,\pm}\rangle = \frac{E_{\downarrow,\pm}^N}{\omega}|N_{\downarrow,\pm}\rangle. \tag{B.4}$$

By completing the square:

$$\left[(\hat{a}^\dagger + \alpha_\pm)(\hat{a} + \alpha_\pm)\right]|N_{\uparrow,\pm}\rangle = \left(\frac{E^N_{\uparrow,\pm}}{\omega} + \alpha_\pm^2\right)|N_{\uparrow,\pm}\rangle\,, \tag{B.5}$$

$$\left[(\hat{a}^\dagger - \alpha_\mp)(\hat{a} - \alpha_\mp)\right]|N_{\downarrow,\pm}\rangle = \left(\frac{E^N_{\downarrow,\pm}}{\omega} + \alpha_\mp^2\right)|N_{\downarrow,\pm}\rangle\,, \tag{B.6}$$

where we defined $\alpha_\pm = (g_1 \pm g_2)/\omega$. Taking $\omega, g_1, g_2$ to be real, the left-hand side can be rewritten in terms of displacement operators $\hat{D}(\alpha) = \exp[\alpha(\hat{a}^\dagger - \hat{a})]$:

$$\left[(\hat{a}^\dagger + \alpha_\pm)(\hat{a} + \alpha_\pm)\right]|N_{\uparrow,\pm}\rangle = \hat{D}^\dagger(\alpha_\pm)\hat{a}^\dagger\hat{a}\hat{D}(\alpha_\pm)|N_{\uparrow,\pm}\rangle\,,$$
$$\left[(\hat{a}^\dagger - \alpha_\mp)(\hat{a} - \alpha_\mp)\right]|N_{\downarrow,\pm}\rangle = \hat{D}^\dagger(-\alpha_\mp)\hat{a}^\dagger\hat{a}\hat{D}(-\alpha_\mp)|N_{\downarrow,\pm}\rangle\,. \tag{B.7}$$

The new eigenstates are displaced Fock number states:

$$|N^N_{\uparrow,\pm}\rangle = \hat{D}^\dagger\left(\frac{g_1 \pm g_2}{\omega}\right)|N\rangle \equiv |N_{\uparrow,\pm}\rangle\,, \tag{B.8}$$

$$|N^N_{\downarrow,\pm}\rangle = \hat{D}^\dagger\left(-\frac{g_1 \mp g_2}{\omega}\right)|N\rangle \equiv |N_{\downarrow,\pm}\rangle\,, \tag{B.9}$$

with eigenvalues

$$E^N_{\uparrow,+} = E^N_{\downarrow,-} = \omega\left[N - \frac{(g_1 + g_2)^2}{\omega^2}\right]\,, \tag{B.10}$$

$$E^N_{\uparrow,-} = E^N_{\downarrow,+} = \omega\left[N - \frac{(g_1 - g_2)^2}{\omega^2}\right]\,. \tag{B.11}$$

## B.2 Qutrit case

The eigenstates $|m\rangle$ of the qutrit operator $(\hat{J}_3^+ + \hat{J}_3^-)$ with eigenvalues $E_m = 0, \pm 2$, can be already obtained through diagonalization: $|0\rangle = \frac{1}{2}(-\sqrt{2}, 0, \sqrt{2})^T$, $|+\rangle = \frac{1}{2}(1, \sqrt{2}, 1)^T$, $|-\rangle = \frac{1}{2}(1, -\sqrt{2}, 1)^T$ in the qutrit number basis $\{|0\rangle, |1\rangle, |2\rangle\}$.

The eigenstates $|N_{\sigma,m}\rangle$ satisfy the set of six eigenvalue equations:

$$\left[\hat{a}^\dagger\hat{a} \pm \frac{g_1}{\omega}(\hat{a}^\dagger + \hat{a})\right]|N_{\uparrow\downarrow,0}\rangle = \frac{E^N_{\uparrow\downarrow,0}}{\omega}|N_{\uparrow\downarrow,0}\rangle\,, \tag{B.12}$$

$$\left[\hat{a}^\dagger\hat{a} + \frac{(g_1 \pm 2g_2)}{\omega}(\hat{a}^\dagger + \hat{a})\right]|N_{\uparrow,\pm}\rangle = \frac{E^N_{\uparrow,\pm}}{\omega}|N_{\uparrow,\pm}\rangle\,, \tag{B.13}$$

$$\left[\hat{a}^\dagger\hat{a} - \frac{(g_1 \pm 2g_2)}{\omega}(\hat{a}^\dagger + \hat{a})\right]|N_{\downarrow,\pm}\rangle = \frac{E^N_{\uparrow,\pm}}{\omega}|N_{\downarrow,\pm}\rangle\,. \tag{B.14}$$

Repeating the steps of the previous section, we obtain the eigenstates:

$$|N^N_{\uparrow\downarrow,0}\rangle = \hat{D}^\dagger\left(\pm\frac{g_1}{\omega}\right)|N\rangle \equiv |N_{\uparrow\downarrow,0}\rangle\,, \tag{B.15}$$

$$|N^N_{\uparrow,\pm}\rangle = \hat{D}^\dagger\left(\frac{g_1 \pm 2g_2}{\omega}\right)|N\rangle \equiv |N_{\uparrow,\pm}\rangle\,, \tag{B.16}$$

$$|N^N_{\downarrow,\pm}\rangle = \hat{D}^\dagger\left(-\frac{g_1 \mp 2g_2}{\omega}\right)|N\rangle \equiv |N_{\downarrow,\pm}\rangle\,, \tag{B.17}$$

with eigenvalues

$$E^N_{\uparrow\downarrow,0} = \omega\left[N - \frac{g_1^2}{\omega^2}\right], \tag{B.18}$$

$$E^N_{\uparrow,+} = E^N_{\downarrow,-} = \omega\left[N - \frac{(g_1 + 2g_2)^2}{\omega^2}\right], \tag{B.19}$$

$$E^N_{\uparrow,-} = E^N_{\downarrow,+} = \omega\left[N - \frac{(g_1 - 2g_2)^2}{\omega^2}\right]. \tag{B.20}$$

## B.3 Ququart case

The eigenstates $|m\rangle$ of the ququart operator $(\hat{J}_4^+ + \hat{J}_4^-)$ are obtained as:
$|a\rangle = \frac{1}{4}(-\sqrt{2}, \sqrt{6}, -\sqrt{6}, \sqrt{2})^T$, $|b\rangle = \frac{1}{4}(\sqrt{6}, -\sqrt{2}, -\sqrt{2}, \sqrt{6})^T$, $|c\rangle = \frac{1}{4}(-\sqrt{6}, -\sqrt{2}, \sqrt{2}, \sqrt{6})^T$,
$|d\rangle = \frac{1}{4}(\sqrt{2}, \sqrt{6}, \sqrt{6}, \sqrt{2})^T$, with eigenvalues $E_{a,b,c,d} = \{-3, -1, 1, 3\}$.

Repeating the steps of the previous section, we obtain the eigenstates:

$$|N^N_{\uparrow,d}\rangle = \hat{D}^\dagger\left(\frac{g_1 + 3g_2}{\omega}\right)|N\rangle, \tag{B.21}$$

$$|N^N_{\downarrow,a}\rangle = \hat{D}^\dagger\left(-\frac{g_1 + 3g_2}{\omega}\right)|N\rangle, \tag{B.22}$$

$$|N^N_{\uparrow,c}\rangle = \hat{D}^\dagger\left(\frac{g_1 + g_2}{\omega}\right)|N\rangle, \tag{B.23}$$

$$|N^N_{\downarrow,b}\rangle = \hat{D}^\dagger\left(-\frac{g_1 + g_2}{\omega}\right)|N\rangle, \tag{B.24}$$

$$|N^N_{\uparrow,b}\rangle = \hat{D}^\dagger\left(\frac{g_1 - g_2}{\omega}\right)|N\rangle, \tag{B.25}$$

$$|N^N_{\downarrow,c}\rangle = \hat{D}^\dagger\left(-\frac{g_1 - g_2}{\omega}\right)|N\rangle, \tag{B.26}$$

$$|N^N_{\uparrow,a}\rangle = \hat{D}^\dagger\left(\frac{g_1 - 3g_2}{\omega}\right)|N\rangle, \tag{B.27}$$

$$|N^N_{\downarrow,d}\rangle = \hat{D}^\dagger\left(-\frac{g_1 - 3g_2}{\omega}\right)|N\rangle, \tag{B.28}$$

with eigenvalues

$$E^N_{\uparrow,d} = E^N_{\downarrow,a} = \omega\left[N - \frac{(g_1 + 3g_2)^2}{\omega^2}\right], \tag{B.29}$$

$$E^N_{\uparrow,c} = E^N_{\downarrow,b} = \omega\left[N - \frac{(g_1 + g_2)^2}{\omega^2}\right], \tag{B.30}$$

$$E^N_{\uparrow,b} = E^N_{\downarrow,c} = \omega\left[N - \frac{(g_1 - g_2)^2}{\omega^2}\right], \tag{B.31}$$

$$E^N_{\uparrow,a} = E^N_{\downarrow,d} = \omega\left[N - \frac{(g_1 - 3g_2)^2}{\omega^2}\right]. \tag{B.32}$$

## B.4 Qudit case

The above explicit results for the eigenvalues and eigenstates can be also obtain in the general qudit case by applying two unitary transformations to the adiabatic limit Hamiltonian in

Eq. (B.1). The transformations are of the Lang-Firsov type [56]:

$$\hat{U}_1 = e^{(g_1/\omega)\hat{\sigma}_x(\hat{a}^\dagger - \hat{a})}, \tag{B.33}$$

$$\hat{U}_2 = e^{(g_2/\omega)(\hat{J}_d^+ + \hat{J}_d^-)(\hat{a}^\dagger - \hat{a})}. \tag{B.34}$$

The transformed Hamiltonian can be written as

$$\tilde{H} = \omega\hat{a}^\dagger\hat{a} - \omega\left[\frac{g_2}{\omega}\left(\hat{J}_d^+ + \hat{J}_d^-\right) + \frac{g_1}{\omega}\hat{\sigma}_x\right]^2. \tag{B.35}$$

The eigenvalues are found immediately from those of of $\hat{\sigma}_x$, $\hat{J}_d^+ + \hat{J}_d^-$, and $\hat{a}^\dagger\hat{a}$, namely $\sigma = \pm 1$, $m = -d, -d+2, \ldots, d-2, d$, and $N = 0, 1, \ldots$, respectively. The eigenstates can be correspondingly given as $|\sigma, m, N\rangle$, where after the unitary transformations the oscillator Fock state is independent of the qubit and qudit state. In the original basis, the eigenstates are found by applying to these eigenstates the above unitary transformations which, after acting on eigenstates of $\hat{\sigma}_x$ and $\hat{J}_d^+ + \hat{J}_d^-$, reduce to displacement operators acting on the oscillator states, with the displacement depending on $\sigma$ and $m$.

## C  Perturbation correction to the energy and state

We are interested in resolving the degeneracy in the subspace (denoted by $D$) spanned by $\{|\uparrow, +, 0_{\uparrow,+}\rangle, |\downarrow, -, 0_{\downarrow,-}\rangle\}$ with degenerate energy $E^0_{\uparrow,+} = E^0_{\downarrow,-}$.

The first order perturbation equation reads [57]:

$$\hat{H}_0|\psi_n^{(1)}\rangle + \hat{H}'|\psi_n^{(0)}\rangle = E_n^{(0)}|\psi_n^{(1)}\rangle + E_n^{(1)}|\psi_n^{(0)}\rangle, \tag{C.1}$$

and first order energy correction:

$$E_n^{(1)} = \langle\psi_n^{(0)}|\hat{H}'|\psi_n^{(0)}\rangle. \tag{C.2}$$

As we show below, all the first order corrections are zero. Since the energy degeneracy in the subspace of interest $D$ is not lifted by first order corrections, we need to find another good basis that diagonalize a new matrix $M$:

$$\langle i|\hat{M}|j\rangle = \sum_{k \notin D}\frac{\langle i|\hat{H}'|k\rangle\langle k|\hat{H}'|j\rangle}{E_n^{(0)} - E_k^{(0)}}, \tag{C.3}$$

which turns out to be a $2 \times 2$ symmetric matrix, with $M_{00} = M_{11}$ and $M_{01} = M_{10}$. Diagonalizing the matrix $\hat{M}$, we obtain the new basis:

$$|\psi_\pm^{(0)}\rangle = \frac{1}{\sqrt{2}}\left(|\uparrow, +, 0_{\uparrow,+}\rangle \pm |\downarrow, -, 0_{\downarrow,-}\rangle\right). \tag{C.4}$$

The energy degeneracy in the subspace with lowest energy is lifted at the second order in perturbation theory. Therefore, the degeneracy-lifted energies are given by

$$\mathcal{E}_\pm = E_n^{(0)} + E_n^{(2)} = E_n^{(0)} + M_{00} \pm M_{01}. \tag{C.5}$$

The first order correction to the ground state are obtained through the formula

$$|\psi_\pm^{(1)}\rangle = \sum_{m \notin D}\frac{\langle m|\hat{H}'|\psi_\pm^{(0)}\rangle}{E_n^{(0)} - E_m^{(0)}}, \tag{C.6}$$

which produces the higher order terms in Eq. (12).

## C.1 Qubit case

The Hamiltonian $\hat{H}_{d=2}$ in the $|\sigma, m, N_{\sigma,m}\rangle$ basis with row and column order $|\uparrow, +, 0_{\uparrow,+}\rangle$, $|\downarrow, -, 0_{\downarrow,-}\rangle$, $|\uparrow, -, 0_{\uparrow,-}\rangle$, $|\downarrow, +, 0_{\downarrow,+}\rangle$, ... reads:

$$\hat{H}_{d=2} = \hat{H}_0 + \hat{H}' = \begin{bmatrix} E^0_{\uparrow,+} & 0 & 0 & 0 & \dots \\ 0 & E^0_{\downarrow,-} & 0 & 0 & \dots \\ 0 & 0 & E^0_{\uparrow,-} & 0 & \dots \\ 0 & 0 & 0 & E^0_{\downarrow,+} & \dots \\ \vdots & \vdots & \vdots & \vdots & \vdots \end{bmatrix} + \begin{bmatrix} 0 & 0 & t & u & \dots \\ 0 & 0 & u & t & \dots \\ t & u & 0 & 0 & \dots \\ u & t & 0 & 0 & \dots \\ \vdots & \vdots & \vdots & \vdots & \ddots \end{bmatrix}, \tag{C.7}$$

where we have defined

$$t = -\frac{1}{2}\Omega_2 \langle 0_{\uparrow,+}|0_{\uparrow,-}\rangle, \quad u = -\frac{1}{2}\Omega_1 \langle 0_{\uparrow,+}|0_{\downarrow,+}\rangle, \tag{C.8}$$

and used the symmetry of the coherent states overlaps, which are real numbers in our case, reading

$$\langle 0_{\uparrow,+}|0_{\uparrow,-}\rangle = \langle 0_{\downarrow,-}|0_{\downarrow,+}\rangle = \exp(-2g_2^2/\omega^2),$$
$$\langle 0_{\uparrow,+}|0_{\downarrow,+}\rangle = \langle 0_{\downarrow,-}|0_{\uparrow,-}\rangle = \exp(-2g_1^2/\omega^2). \tag{C.9}$$

Using Eq. (C.3), we can compute the matrix elements of $M$:

$$M_{00} = M_{11} = -\frac{\omega}{16g_1g_2}(\Omega_1^2 e^{-4g_1^2/\omega^2} + \Omega_2^2 e^{-4g_2^2/\omega^2}), \tag{C.10}$$

$$M_{01} = M_{10} = \frac{\omega\Omega_1\Omega_2}{8g_1g_2}e^{-2(g_1^2+g_2^2)/\omega^2}, \tag{C.11}$$

and the second order energy corrections

$$E_n^{(2)} = -\frac{\omega^2}{16g_1g_2}(\Omega_1^2 e^{-4g_1^2/\omega^2} + \Omega_2^2 e^{-4g_2^2/\omega^2}) \pm \frac{\omega^2\Omega_1\Omega_2}{8g_1g_2}e^{-2(g_1^2+g_2^2)/\omega^2}. \tag{C.12}$$

## C.2 Qutrit case

We proceed as in the previous section and write $\hat{H}_{d=3}$ in the ordered basis $|\uparrow, 0, 0_{\uparrow,0}\rangle$, $|\downarrow, 0, 0_{\downarrow,0}\rangle$, $|\uparrow, +, 0_{\uparrow,+}\rangle$, $|\downarrow, -, 0_{\downarrow,-}\rangle$, $|\uparrow, -, 0_{\uparrow,-}\rangle$, $|\downarrow, +, 0_{\downarrow,+}\rangle$, ...:

$$\hat{H}_{d=3} = \hat{H}_0 + \hat{H}' = \begin{bmatrix} E_1 & 0 & 0 & 0 & 0 & 0 & \dots \\ 0 & E_1 & 0 & 0 & 0 & 0 & \dots \\ 0 & 0 & E_0 & 0 & 0 & 0 & \dots \\ 0 & 0 & 0 & E_0 & 0 & 0 & \dots \\ 0 & 0 & 0 & 0 & E_2 & 0 & \dots \\ 0 & 0 & 0 & 0 & 0 & E_2 & \dots \\ \vdots & \vdots & \vdots & \vdots & \vdots & \vdots & \ddots \end{bmatrix} + \begin{bmatrix} 0 & u & t & 0 & t & 0 & \dots \\ u & 0 & 0 & t & 0 & t & \dots \\ t & 0 & 0 & 0 & 0 & u & \dots \\ 0 & t & 0 & 0 & u & 0 & \dots \\ t & 0 & 0 & u & 0 & 0 & \dots \\ 0 & t & u & 0 & 0 & 0 & \dots \\ \vdots & \vdots & \vdots & \vdots & \vdots & \vdots & \ddots \end{bmatrix}, \tag{C.13}$$

$$\tag{C.14}$$

where we defined

$$E_0 = E^0_{\uparrow,+}, \quad E_1 = E^0_{\uparrow,0}, \quad E_2 = E^0_{\uparrow,-},$$

$$t = -\frac{\sqrt{2}}{2}\Omega_2 \exp(-2g_2^2/\omega^2), u = -\frac{1}{2}\Omega_1 \exp(-2g_1^2/\omega^2). \tag{C.15}$$

The symmetric matrix $M$ of Eq. (C.3) reads:

$$M_{00} = M_{11} = -\frac{\omega\Omega_1^2}{32g_1g_2}e^{-4g_1^2/\omega^2} - \frac{\omega\Omega_2^2}{8(g_2^2 + g_1g_2)}e^{-4g_2^2/\omega^2},$$

$$M_{01} = M_{10} = 0. \tag{C.16}$$

Since the off-diagonal elements of the matrix $M$ are zero, the degeneration is not resolved at second order in perturbation theory, and we need to find higher order perturbative contributions from the Hamiltonian Eq. (C.13). We show here a simplified procedure to obtain the lowest order correction which resolves the degeneracy, without using the general expression, which is pretty involved.

The crucial observation is that the matrix can be exactly diagonalized for $t = 0$, with three independent rotations in the three subspaces (generated by the base vectors) $D_1 = \{|\uparrow,0,0_{\uparrow,0}\rangle, |\downarrow,0,0_{\downarrow,0}\rangle\}$, $D_0 = \{|\uparrow,+,0_{\uparrow,+}\rangle, |\downarrow,+,0_{\downarrow,+}\rangle\}$, $D_2 = \{|\downarrow,-,0_{\downarrow,-}\rangle, |\uparrow,-,0_{\uparrow,-}\rangle\}$. More precisely, the $D_1$ subspace is non-degenerate, with eigenvalues $E_1 \pm u$. On the other hand, the energy corrections in the $D_0, D_2$ subspaces are of order $u^2$, and the degeneration is not removed. More precisely, keeping only leading orders in $u$, we have $E_0 \to \tilde{E}_0 = E_0 + \frac{u^2}{E_0-E_2}$ and $E_2 \to \tilde{E}_2 = E_2 + \frac{u^2}{E_2-E_0}$. The ordered basis where $\hat{H}_{d=3}(t=0)$ is diagonal reads

$$\frac{1}{\sqrt{2}}(|\uparrow,0,0_{\uparrow,0}\rangle + |\downarrow,0,0_{\downarrow,0}\rangle), \frac{1}{\sqrt{2}}(|\uparrow,0,0_{\uparrow,0}\rangle - |\downarrow,0,0_{\downarrow,0}\rangle),$$

$$\left[1 - \frac{u^2}{2(E_0-E_2)^2}\right]|\uparrow,+,0_{\uparrow,+}\rangle + \frac{u}{E_0-E_2}|\uparrow,-,0_{\uparrow,-}\rangle,$$

$$\left[1 - \frac{u^2}{2(E_0-E_2)^2}\right]|\downarrow,-,0_{\downarrow,-}\rangle + \frac{u}{E_0-E_2}|\downarrow,+,0_{\downarrow,+}\rangle,$$

$$\frac{u}{E_0-E_2}|\downarrow,-,0_{\downarrow,-}\rangle + \left[1 - \frac{u^2}{2(E_0-E_2)^2}\right]|\downarrow,+,0_{\downarrow,+}\rangle,$$

$$\frac{u}{E_0-E_2}|\uparrow,+,0_{\uparrow,+}\rangle + \left[1 - \frac{u^2}{2(E_0-E_2)^2}\right]|\uparrow,-,0_{\uparrow,-}\rangle. \tag{C.17}$$

Note that the ground state subspace with energy $\tilde{E}_0$ differs from the product state basis only by terms of order $u$.

Since the Hamiltonian can be exactly diagonalized with $t = 0$, we can evaluate the leading contribution by treating $H' = \hat{H}_{d=3} - \hat{H}_{d=3}(t=0)$ as perturbation of $\hat{H}_{d=3}(t=0)$. Namely, we rewrite the Hamiltonian in the previous basis, reading (keeping terms up to $u^2$)

$$\hat{H}_{d=3} = \begin{bmatrix} E_1+u & 0 & t_+ & t_+ & t_- & t_- & \dots \\ 0 & E_1-u & t_+ & t_- & t_- & t_+ & \dots \\ t_+ & t_+ & \tilde{E}_0 & 0 & 0 & 0 & \dots \\ t_+ & t_- & 0 & \tilde{E}_0 & 0 & 0 & \dots \\ t_- & t_- & 0 & 0 & \tilde{E}_2 & 0 & \dots \\ t_- & t_+ & 0 & 0 & 0 & \tilde{E}_2 & \dots \\ \vdots & \vdots & \vdots & \vdots & \vdots & \vdots & \ddots \end{bmatrix}, \tag{C.18}$$

where

$$t_{\pm} = \frac{-tu}{E_0 - E_2} \pm \left[ \frac{t}{\sqrt{2}} - \frac{tu^2}{2\sqrt{2}(E_0 - E_2)^2} \right].$$ (C.19)

Since $t$ is not explicitly present in the diagonal of the rotated matrix, the first order correction is zero. The second order correction in $t$ can be computed with degenerate second order perturbation theory, giving a $2 \times 2$ symmetric matrix $\tilde{M}$, with

$$\tilde{M}_{00} = \tilde{M}_{11} = \frac{t^2}{E_0 - E_1} + \frac{2E_0 - E_1 - E_2}{(E_0 - E_2)(E_0 - E_1)^3} t^2 u^2,$$

$$\tilde{M}_{01} = \tilde{M}_{10} = \frac{3E_0 - 2E_1 - E_2}{(E_0 - E_1)^2(E_0 - E_2)} t^2 u.$$ (C.20)

This implies that the degeneration is lifted at the third order in perturbation theory. The basis which diagonalize the matrix $\tilde{M}$ differs from the GHZ state Eq. (13) only by terms of order $u$, as discussed in the main text.

## C.3 Ququart case

We write $\hat{H}_{d=4}$ in the basis: $|\uparrow, d, 0_{\uparrow,d}\rangle, |\downarrow, a, 0_{\downarrow,a}\rangle, |\uparrow, c, 0_{\uparrow,c}\rangle, |\downarrow, b, 0_{\downarrow,b}\rangle, |\uparrow, b, 0_{\uparrow,b}\rangle, |\downarrow, c, 0_{\downarrow,c}\rangle,$
$|\uparrow, a, 0_{\uparrow,a}\rangle, |\downarrow, d, 0_{\downarrow,d}\rangle, \ldots$:

$$
\hat{H}_{d=4} = \hat{H}_0 + \hat{H}'
$$

$$
= \begin{bmatrix}
E_0 & 0 & 0 & 0 & 0 & 0 & 0 & 0 & \cdots \\
0 & E_0 & 0 & 0 & 0 & 0 & 0 & 0 & \cdots \\
0 & 0 & E_1 & 0 & 0 & 0 & 0 & 0 & \cdots \\
0 & 0 & 0 & E_1 & 0 & 0 & 0 & 0 & \cdots \\
0 & 0 & 0 & 0 & E_2 & 0 & 0 & 0 & \cdots \\
0 & 0 & 0 & 0 & 0 & E_2 & 0 & 0 & \cdots \\
0 & 0 & 0 & 0 & 0 & 0 & E_3 & 0 & \cdots \\
0 & 0 & 0 & 0 & 0 & 0 & 0 & E_3 & \cdots \\
\vdots & \vdots & \vdots & \vdots & \vdots & \vdots & \vdots & \vdots & \ddots
\end{bmatrix}
+ \begin{bmatrix}
0 & 0 & t & 0 & 0 & 0 & 0 & u & \cdots \\
0 & 0 & 0 & t & 0 & 0 & u & 0 & \cdots \\
t & 0 & 0 & 0 & v & u & 0 & 0 & \cdots \\
0 & t & 0 & 0 & u & v & 0 & 0 & \cdots \\
0 & 0 & v & u & 0 & 0 & t & 0 & \cdots \\
0 & 0 & u & v & 0 & 0 & 0 & t & \cdots \\
0 & u & 0 & 0 & t & 0 & 0 & 0 & \cdots \\
u & 0 & 0 & 0 & 0 & t & 0 & 0 & \cdots \\
\vdots & \vdots & \vdots & \vdots & \vdots & \vdots & \vdots & \vdots & \ddots
\end{bmatrix},
$$
(C.21)

where we define:

$$E_0 = E^0_{\uparrow,d}, \quad E_1 = E^0_{\uparrow,c}, \quad E_2 = E^0_{\uparrow,b}, \quad E_3 = E^0_{\uparrow,a},$$

$$t = \frac{\sqrt{3}}{2}\Omega_2 e^{-\frac{2g_2^2}{\omega^2}}, \quad u = -\frac{1}{2}\Omega_1 e^{-\frac{2g_1^2}{\omega^2}}, \quad v = \Omega_2 e^{-\frac{2g_2^2}{\omega^2}}.$$ (C.22)

The matrix elements of the symmetric matrix $M$ are given by:

$$M_{00} = M_{11} = -\frac{\omega \Omega_1^2}{48 g_1 g_2} e^{-\frac{4g_1^2}{\omega^2}} - \frac{3\omega \Omega_2^2}{4(8g_2^2 + 4g_1 g_2)} e^{-\frac{4g_2^2}{\omega^2}},$$

$$M_{01} = M_{10} = 0.$$ (C.23)

As in the qutrit case, the energy degeneration is not lifted at second order in perturbation theory. The procedure for the determination of higher order corrections discussed in the previous subsection can be generalized to the ququart case.

   We first consider the matrix (C.21) for $t = 0$. The matrix $\hat{H}_{d=4}(t = 0)$ is composed of two orthogonal subspaces of dimension 4, which can be exactly diagonalized. The first subspace $D_{03}$, spanned by the vectors $\{|\uparrow, d, 0_{\uparrow,d}\rangle, |\downarrow, a, 0_{\downarrow,a}\rangle, |\uparrow, a, 0_{\uparrow,a}\rangle, |\downarrow, d, 0_{\downarrow,d}\rangle\}\}$ is characterized by two doubly-degenerate eigenvalues (here written at leading orders in $u, v$) where

$E_0 \to \tilde{E}_0 = E_0 + u^2/(E_0 - E_3)$, $E_3 \to \tilde{E}_3 = E_3 + u^2/(E_3 - E_0)$. The eigenvalues of the ortogonal subspace $D_{12}$, spanned by the vectors $\{|\uparrow, c, 0_{\uparrow,c}\rangle, |\downarrow, b, 0_{\downarrow,b}\rangle, |\uparrow, b, 0_{\uparrow,b}\rangle, |\downarrow, c, 0_{\downarrow,c}\rangle\}$, are non degenerate and read $E_1 \to \tilde{E}_{1,\pm} = E_1 + (u \pm v)^2/(E_1 - E_2)$, $E_2 \to \tilde{E}_{2,\pm} = E_2 - (u \pm v)^2/(E_1 - E_2)$.

Then, we reintroduce the dependence on $t$ through perturbative expansion in the operator $H' = \hat{H}_{d=4} - \hat{H}_{d=4}(t=0)$. As in the qutrit case the first order correction in $t$ is zero. The second order corrections in $t$ are computed again with degenerate second order perturbation theory. The $2 \times 2$ symmetric matrix $\tilde{M}$ reads

$$
\begin{aligned}
\tilde{M}_{00} = \tilde{M}_{11} &= \frac{t^2}{E_0 - E_1} + \frac{t^2 v^2}{(E_0 - E_1)^2(E_0 - E_2)} + \frac{t^2 u^2(2E_0 - E_1 - E_3)(E_0 + E_2 - E_1 - E_3)}{(E_0 - E_1)^2(E_0 - E_2)(E_0 - E_3)^2}, \\
\tilde{M}_{01} = \tilde{M}_{10} &= \frac{2(2E_0 - E_1 - E_3)t^2 uv}{(E_0 - E_1)^2(E_0 - E_2)(E_0 - E_3)}.
\end{aligned}
\tag{C.24}
$$

Hence, the degeneration of the ground state is removed at the fourth order in perturbation theory. With a similar expansion in the basis which diagonalize the matrix $\hat{H}_{d=4}(t=0)$ (not shown here), one notes that the basis which diagonalize the matrix $\tilde{M}$ differs from the GHZ state Eq. (13) only by terms of order $u, v, t$.

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
