# Peer review of "GHZ-like states in the Qubit-Qudit Rabi Model"

_SciPost Physics, doi:SciPost Phys. 11, 099 (2021)_

## Round 2 · Referee Report · Anonymous (Referee 1) · 2021-7-25

Strengths

1- Timely topic with results relevant for quantum technology 2- Results appear to satisfy all the general criteria for SciPost Physics 3- Expanded discussion on the dynamics sections

Weaknesses

1- None. Paper ready for publication in my view.

Report

The authors have addressed all my comments from my first report. The new version provides more insight into the dynamical aspect of the Qubit-Qudit Rabi Model. I recommend publication in its present form.

Requested changes

None

---

## Round 2 · Referee Report · Anonymous (Referee 2) · 2021-10-21

Strengths

– Clearly written
– Reports interesting results with possible applicaitons for generating highly entangled states

Weaknesses

– The notation used is sometimes complicated

Report

The authors of this paper consider a two-level system (qubit) coupled to a multi-level system (qudit) through a common resonator, for example as can be realized in cavity QED. They obtain analytical solutions through weak- and strong-coupling perturbation theory and find that the strong-coupling ground state of the system is of the GHZ type. They verify these analytical solutions by comparing to exact numerical solutions for qudits with two, three, and four levels. They also study the nature of entanglement in the strong-coupling regime quantified by the nagativity and find that the bipartite entanglement between the qubit and qudit is suppressed at strong coupling. Finally they study both quench and adiabatic dynamics of the system and show that the GHZ state can be prepared adiabatically by switching on the couplings.

This work presents a clear advance in the field of quantum information in hybrid quantum systems showing a way to prepare and control highly entangled quantum states. The paper is well written and the results are presented clearly and convincingly. The notation used can get a bit too complex at times, but I think the authors have made sufficient notes to clarify their use. As such, I recommend the paper to be published in SciPost Physics.

I did notice a couple of small typos that should be corrected before final publication:

-- Typo in the abstract: "The ground state of the strongly coupled system is a found of ..." should be "The ground state of the strongly coupled system is found to be of ..."

-- P. 4 first paragraph, starting with "In the low coupling regime, the
analytical expression of Eq. (9) gives ..." the frequencies appear as upper-case $\Omega$, but I think they should be lower-case $\omega$ as "$dg_1 \leq 0.4\omega$" and "$dg_1 \leq 0.3\omega$".

---

## Round 2 · List of Changes

1- The discussion of the dynamics has been significantly extended, following the comment of the Referee. In particular, we inserted a new figure and included new plots, computing numerically the fast Fourier transform of the time evolved signals (both for the analytics and the numerics). We derived a new equation (Eq.18). The definition of $\tilde H$ is given in a new equation (Eq.10, previously defined as in-line equation).
2- We extended the discussion in the conclusions. More precisely, we discuss possible platforms for the experimental implementation of our model, and future perspectives, addressing the comments of the Referee.
3- We inserted new references: [27] Rev. Mod. Phys.91, 025005 (2019), [51] Phys. Rev. Lett. 105, 023601 (2010), [52] New J. Phys. 19, 023022 (2017), [53] J. Phys. A: Math. Theor. 50 294001 (2017), [54] Phys. Rev. Lett. 105, 237001 (2010), [55] Nat. Phys. 13, 39 (2017), [56] arXiv:2106.01669 [quant-ph], [57] arXiv:2104.03045 [cond-mat.mes-hall], [58] Nat. Commun.10, 3011 (2019), [59] arXiv:2104.14490 [quant-ph].

---

## Editorial Decision

published